# Parylene-Sealed Perovskite Nanocrystals Down-Shifting Layer for Luminescent Spectral Matching in Thin Film Photovoltaics

**DOI:** 10.3390/nano13010210

**Published:** 2023-01-03

**Authors:** Ana Pinheiro, Andreia Ruivo, João Rocha, Marta Ferro, Joana Vaz Pinto, Jonas Deuermeier, Tiago Mateus, Ana Santa, Manuel J. Mendes, Rodrigo Martins, Sandra Gago, César A. T. Laia, Hugo Águas

**Affiliations:** 1R&D Unit VICARTE—Vidro e Cerâmica para as Artes, NOVA School of Science and Technology, Largo da Torre, 2825-516 Caparica, Portugal; 2Associated Laboratory for Green Chemistry (LAQV) of the Network of Chemistry and Technology (REQUIMTE), NOVA School of Sciences and Technology, FCT NOVA, Largo da Torre, 2825-516 Caparica, Portugal; 3Department of Chemistry, CICECO-Aveiro Institute of Materials, University of Aveiro, 3810-193 Aveiro, Portugal; 4Department of Materials and Ceramic Engineering, CICECO-Aveiro Institute of Materials, University of Aveiro, 3810-193 Aveiro, Portugal; 5CENIMAT|i3N, NOVA School of Sciences and Technology and CEMOP/UNINOVA, FCT NOVA, Largo da Torre, 2825-516 Caparica, Portugal

**Keywords:** photovoltaics, perovskite nanocrystals, luminescent materials, parylene type C, silicon solar cells

## Abstract

The present contribution aims to enhance solar cells’ performance via the development of advanced luminescent down-shifting based on encapsulated nanostructured perovskite materials. Here, thin films of inorganic lead halide (CsPbBr_3_) perovskite nanocrystal luminophores were synthetized, by hot-injection, deposited on glass substrates by spin-coating, and encapsulated with parylene type C, via chemical vapor deposition, to protect and stabilize the films. The optical properties of these thin films were characterized by absorption, emission and 2D contour spectra, their structure by X-ray diffraction and X-ray photoelectron spectroscopy, and the morphology by Scanning Transmission Electron microscopy. I–V curve and spectral response nanocrystalline silicon photovoltaic (nc-Si:H PV) cells were studied in the absence and presence of the perovskite and parylene luminescent down-shifting layers. The incorporation of the CsPbBr_3_ nanocrystals and their encapsulation with the parylene type C polymeric coating led to an increase in the current generated and the spectral response of the PV cells in the regime of the nanocrystals’ fluorescence emission. A 3.1% increase in the short circuit current density and a 5.6% increase in the power conversion efficiency were observed.

## 1. Introduction

Silicon based photovoltaic cells have been extensively explored and have become the dominant photovoltaic systems. Silicon is a widely abundant and non-pollutant element, reaching, in crystalline silicon solar cells, very good efficiency, up to 26.8% [1,2,3]. Nevertheless, silicon based solar cells, including nanocrystalline silicon solar cells, display low spectral response for high energy photons (in the blue and UV range), which are not converted by the device and raise thermalization issues [4]. To overcome this drawback, a luminescent down-shifting (LSD) layer may be included in the system. This allows the absorption of the high energy photons of the solar spectrum via a third-party luminophore (e.g., dye, lanthanide, or nanocrystal), followed by photon emission at lower energies (closer to band gap), which are more efficiently converted by the Si solar cells. This process increases current generation and power conversion efficiency, improving the overall performance of the solar cell [5]. Furthermore, the possibility of coupling LSD layers with the solar cells in a planar configuration, i.e., depositing them on top of the as-fabricated solar cells, simplifies their incorporation without the need for architectural or electrical matching [4].

Some of the LDS layer requirements concern the optical properties of the incorporated luminescent species and the encapsulation material. The luminophores should display a high photoluminescent quantum yield (PLQY) and, simultaneously, have a low overlap between the emission range and the spectral response of the solar component. Moreover, the encapsulating matrix should have a very high light transmittance, especially in the visible range, where the solar cell absorbs the incident light. The matrix should also have a suitable refractive index for good optical matching, allowing a good optical interface transition with the solar cell, and for a homogeneous distribution of the luminescent species along the whole plane [4].

Semiconductor nanocrystals are excellent luminophore candidates to be used in LSD layers. Amid these, inorganic metal halide perovskites have gained much interest due to their excellent optoelectronic properties and wide range applications [6,7,8]. These materials have a crystal structure similar to calcium titanate, CaTiO_3_. They are described by the formula ABX_3_ (X = monovalent Cl^−^, Br^−^, I^−^ anion or a mixture of these halides), where A stands for monovalent species, such as Cs^+^, CH_3_NH_3_^+^ (MA) or CH(NH_2_)_2_^+^ (FA), and B stands for divalent cations, such as Pb^2+^ and Sn^2+^ [8,9]. These compounds show direct and tunable bandgaps, strong light-absorption coefficients, outstanding charge transport properties, and excellent luminescence properties with narrow emission linewidth and high photoluminescence quantum yields [10]. Nevertheless, certain drawbacks of these materials, such as their chemical, photo, and thermal instability, susceptibility to polar solvents, anion-exchange reactions, and moisture and oxygen degradation compromise their application in the field of optoelectronics [6,7,8]. Inorganic lead halide perovskite (e.g., MAPbX_3_ and FAPbX_3_) that are more stable than their organic–inorganic hybrid equivalents have been developed to overcome these issues [7,11]. However, the problems of thermal instability, altering perovskite’s crystalline structure, and moisture and oxygen sensitivity, remain largely unsolved.

The encapsulation of CsPbBr_3_ nanocrystals (NCs) in protective solid matrices is one of the most widely studied approaches for the surface passivation of perovskite materials. Encapsulation may increase the perovskite’s tolerance to humidity and oxygen, preventing degradation, NCs aggregation and anionic exchange between halides, preserving the luminescence properties [6,7,8]. Many polymers are suitable as perovskite NCs encapsulation material as they are transparent, gas and moisture impermeable, mechanically robust and compatible with the perovskite processing conditions, especially concerning temperature limitations (perovskite NCs are very sensitive to temperatures above 150 °C) [7,8]. Among the available polymers, poly-para-xylylene (parylene) stands out. It is an excellent encapsulation material, very stable and impermeable to oxygen and water, with very low surface energy and high mechanical strength. Additionally, it is transparent, its refractive index allows for a gradual optical transition from the air to the TCO layer and, thus, it displays anti-reflective properties, which is an advantage when applied to LSD layers for the improvement of light absorption, by the luminophores and the Si solar cells [12]. There are several studies on the application of different organic polymers, such as polymethyl methacrylate (PMMA), epoxy resin (ER), polystyrene (PS), and polyethylene oxide (PEO), and inorganic materials, such as diamond, silica, glass, ceramic and boron oxide [8]. However, there is a lack of information regarding the application of parylene type C [poly(chloro-p-xylylene)] as a protective coating for the production of CsPbBr_3_ NCs thin films. Parylene type C is the parylene material most effective against moisture vapor transmission and have already been used to passivate thin film devices [7,13]. It also shows the highest tensile and yield strength, no oxidative degeneration and very appealing thermal properties considering the final application [14].

Here, we study the optical properties of the CsPbBr_3_ perovskite NCs thin films produced from the colloidal solution in n-hexane, via spin coating, and encapsulated with parylene type C. We assess the possibility of applying this perovskite and parylene thin films as down-shifting layers to improve the current generation and overall power conversion efficiency (PCE) of nanocrystalline silicon photovoltaic solar cells (nc-Si:H PV solar cells).

## 2. Materials and Methods

Cesium carbonate (Cs_2_CO_3_, Sigma-Aldrich, St. Louis, MO, USA, 99%), lead bromide (PbBr_2,_ Alfa Aesar, Haverhill, MA, USA, 99.9%), oleic acid (OA, Sigma-Aldrich, 90%), 1- octadecene (ODE, Sigma-Aldrich, 90%), oleylamine (OAM, Acros Organics, Waltham, MA, USA, 80–90%), parylene type C (Specialty Coating Systems, Indianapolis, IN, USA) and silane A174 (Sigma-Aldrich) were used as received from commercial sources.

Colloidal solutions of perovskite CsPbBr_3_ NCs were prepared following the hot-injection method reported by Kovalenko et al., with minor modifications [15].

### 2.1. Preparation of Cesium Oleate Precursor

For the synthesis of the NCs, firstly, a cesium-oleate precursor solution was prepared by adding Cs_2_CO_3_ (0.4 g, 1.23 mmol), ODE (15 mL) and OA (1.5 mL of oleic acid) into a 100 mL 2-neck flask. The flask was then degassed under an N_2_ flux, at room temperature, for 20 min, and heated to 150 °C from 1 h 00 to 1 h 30, under stirring. 

### 2.2. Synthesis of CsPbBr_3_ Nanocrystals

PbBr_2_ (g, 0.19 mmol), ODE (5 mL), OA (0.5 mL) and OAM (0.5 mL) were put into a 50 mL 2-neck flask, under an N_2_ flux, at room temperature, for 20 min, and heated to 120 °C for 1 h, under stirring. The temperature was set to 150 °C and the cesium-oleate precursor solution to 110 °C. Then, 0.4 mL of the cesium-oleate solution was quickly added to the PbBr_2_ solution and, 5 to 10 s later still under N_2_ flux, the mixture was put in the ice-water bath and left to cool down. The NCs solution was centrifuged at 5000 rpm for 20 min and the pellet re-disperse in n-hexane.

### 2.3. nc-Si:H Photovoltaic Cells Fabrication

Single junction nanocrystalline silicon (nc-S:H) thin film solar cells in a n-i-p configuration, using the deposition process described by A. Lyubchyk et al. [16], were deposited on glass substrates of 1 mm in thickness and with an area of 10 × 10 cm^2^. The substrate was covered with 50 nm of Aluminum zinc oxide (AZO) to improve the adhesion of the following layers. Aluminum back contacts and a reflective layer of AZO of 120 nm and 75 nm in thickness, respectively, were deposited by PVD methods using a mechanical mask. The silicon layers n-type, intrinsic and p-type layers of 54 nm, 1.5 μm and 25 nm in thickness, respectively, were deposited on a Elettrorava PECVD system, with a substrate temperature of 160 °C. Note that the doped layers were deposited in separated RF-PECVD chambers. The intrinsic absorber layer was deposited on a dedicated deposition chamber using VHF. A layer of TCO (transparent conductive oxide) composed of indium zinc oxide (IZO) of 225 nm in thickness was deposited on top of the total silicon layer by PVD methods, again using a mechanical mask. A final layer of Al with 120 nm was deposited again using mechanical masks next to the IZO which formed the top contact electrical connection pad for characterization. The excess of silicon surrounding the active area of the solar cells was removed using RIE (reactive ion etching) or dry etching. The active area of the solar cells, measured using an optical microscope, is 19.6 mm^2^. The cells were then annealed at 160 °C under vacuum, for 120 min. The deposition parameter can be consulted in Appendix A.

### 2.4. Perovskite Incorporation on Photovoltaic Cells 

Three solar cell replicas were studied (i, ii, and iii). The nc-Si:H solar cells were cleaned with isopropanol prior to the spin-coating of the perovskite thin films. Uniform perovskite thin films were obtained for spin-coating depositions at 1500 rpm for 30 s, with a total deposited volume of 300 μL of a mixture of a 1:2 solution of the re-disperse colloidal solution of NCs and 10% (wt%) polyethylene-hexane solution, from low-density polyethylene ((C_2_H_4_)ₙ, Sigma-Aldrich), respectively, on top of the TCO (IZO) layer of the previously fabricated PV cells. The cell contacts were protected during the deposition and encapsulation processes with Kapton tape. Information on the used PV cells and NCs deposition parameters can be consulted in Appendix A.

### 2.5. Parylene Type C Encapsulation

Parylene was deposited through chemical vapor deposition (CVD) on a PDS 2010 Labcoter 2, Specialty Coating 9 Systems, Inc. 2 g of parylene type C dimer was added to a vaporizer chamber and, after sublimation and pyrolysis of the dimer, the parylene polymerized on the NCs integrated solar cells to obtain a 1 μm encapsulation layer. In this process, an adhesion promoter (silane A174) was added to the chamber. 

### 2.6. Measurements

The absorption spectra were attained using a conventional double beam VARIAN spectrophotometer, model Cary-5000, over the 300–800 nm wavelength range. The emission and excitation 3D profiles, of both the NCs in solution and thin film, were measured on a SPEX ^®^ Fluorolog^®^-3 HORIBA spectrofluorometer (Horiba Scientific, Palaiseau, France), model FL3-22. The emission spectra of the NCs in solution were acquired in the range of 400–750 nm, with a 395 nm excitation wavelength and 1 nm slits. The NCs thin film emission spectra were measured in the 440–550 nm wavelength range and an excitation wavelength of 410 nm, with a 0.5 nm slit. The excitation 3D profiles were only measured for the solid samples in the range 395–600 nm, with an excitation wavelength range of 280–460 nm, with a 0.5 nm slit. Fluorescence decay spectra of the colloidal nanocrystals were acquired on a FLUOROMAX-4 HORIBA spectrofluorometer from Proteomass-Bioscope facility, with excitation wavelength at 460 nm and emission wavelength at 510 nm. The fluorescence quantum yield from the colloidal nanocrystals was measured on a SPEX ^®^ Fluorolog^®^-3 HORIBA spectrofluorometer, model FL3-22, using an integrating sphere with an excitation wavelength at 455 nm, with a 0.8 nm slit, in the range of 465–750 nm. 

The dispersed NCs in solution were also characterized by Scanning Transmission Electron Microscopy (STEM) on a STEM Hitachi HD2700 microscope and by X-Ray Diffraction (XRD) on a PANalyticalX’Pert Pro MDP diffractometer, with a copper anode (Cu Kα radiation, λ = 0.15418 nm) and an 1D X’Celerator detector. Measurements were obtained by continuous scanning in the 10°–90° (2θ) range with a 0.017° step. XRD data was analyzed on a High Score Plus software.

X-ray Photoelectron Spectroscopy was performed on a Kratos Axis Supra spectrometer, with a monochromatized Al K- α radiation running at 225 W and charge neutralization, from electron source only. The binding energies were charge-corrected a posteriori to C 1s at 284.8 eV. The analyzer was set to a pass energy of 10 eV. 

The electrical characterization of the nc-Si:H PV cells was performed by means of I-V curves, measured with an LED Sun Simulator (Newport-Oriel Vera Sol LSH-7520) composed of 19 LEDs at individual wavelengths spaced over the relevant spectrum from 400 to 1100 nm, supplying 1-sun illumination certified AAA in a 51 × 51 mm^2^ area. The spectral response measurements were performed using a Newport-Oriel QuantX-300 system, including a Oriel^®^Instruments software (version REV06). The equipment uses a Newport-Oriel ^®^ 6255, 150 W, Ozone Free Xe ARC Lamp, with a nominal illumination spot size of 0.8 × 1.1 mm, which composes the illumination source of the equipment used. The spectral irradiance profile can be consulted in the method details section of Appendix A.

## 3. Results and Discussion

Colloidal solutions of perovskite CsPbBr_3_ NCs were prepared using the hot-injection method of Kovalenko et al. [15] by re-acting a cesium oleate precursor with a PbBr_2_ solution, in octadecene, with a mixture of oleylamine and oleic acid that enables the solubilization of PbBr_2_ and the colloidal stabilization of NCs. The structure and morphology of these NCs were characterized by XRD and STEM, and the optical properties by absorption and emission spectroscopy. Subsequently, the NCs colloidal solution was deposited on float glass substrates via spin coating (CsPbBr_3_ NCs thin films). NC-parylene thin films were also prepared by deposition of this polymer, with adhesion promoter, on top of CsPbBr_3_ NCs films by chemical vapor deposition. The optical properties of both films were evaluated by absorption and emission spectroscopy and the surface elemental composition assessed by X-Ray Photoelectron Spectroscopy (XPS).

### 3.1. Characterization of the Synthesized NCs

#### 3.1.1. Absorbance and Emission Spectra

NCs show a broad band absorption with a well-defined excitation transition peak at 439 nm, (Figure 1a) and the photoluminescence emission spectrum shows a cyan-green emission with a maximum at 500 nm and a 27.5 nm full-width-at-half-maximum. Together with emission band bell-shape, this indicates that the majority of the colloidal CsPbBr_3_ NCs correspond to a cuboid population, with no contamination of other emissive species. The Stokes shift is 24 nm and the NCs optical bandgap is 2.42 eV, as ascertained from the Tauc plot. The fluorescence quantum yield measured with an integrating sphere is 72%. All these values are in agreement with Kovalenko et al., on CsPbBr_3_ NCs [15].

#### 3.1.2. Photoluminescent Decay

A tri-exponential fitting of the decay measurement in Figure 1b yields an average lifetime of 18.0 ± 0.9 ns, in the range of 1 to 29 ns published by Protesescu et al. [15], but in contrast with the 4.00 ns to 7.62 ns reported for colloidal cubic CsPbBr_3_ [17,18,19,20]. The value 18.0 ± 0.9 ns is close to the values reported for Cs_4_PbBr_6_ NCs, supporting the presence of hexagonal Cs_4_PbBr_6_ NCs, as witnessed by the TEM images in Figure 1d [21].

#### 3.1.3. X-ray Diffraction

The XRD pattern (Figure 1c) shows distinctive reflections that can be attributed to different structures. In particular, two very intense reflections at 15.2° and 30.6° can be attributed to the (101) and (202) reflections of CsPbBr_3_ orthorhombic structure (ICSD: 98-007-7630) [22]. The diffractograms seen in Figure 1c also display other peaks that may indicate the presence of other phases, such as the peak at 37.5°, suggesting the existence of the (112) facet of the CsPbBr_3_ cubic phase (ICSD: 98-007-7631). The presence of (001) and (002) facets of this phase, *Pm*$\bar{3}$*m* space group, cannot be discarded because of the broadness of the peaks [23]. The peaks at 34.1° and 43.7° are attributed to the orthorhombic phase (141) and (242) facets. These results corroborate the STEM evidence (Figure 1d). 

The hexagonal structures with 6.88 Å d-spacing (inset in Figure 1d) are consequence of the (110) Bragg reflections [24]. Comparing the 2.62 and 2.96 values from the sum of the relative difference to the most intense peak (12.9°) from the 12.9°/25.4°/27.5°/28.6° reflections, for the blue asterisks (Figure 1c) and ICSD dataset, respectively, leads to the conclusion that the rhombohedral phase of Cs_4_PbBr_6_ is present. This, together with the hexagonal structures seen by STEM, confirms the presence of the trigonal space group R$\bar{3}$c 0D perovskite Cs_4_PbBr_6_ single crystals.

#### 3.1.4. STEM

Figure 1 d shows three types of NCs differing in size and morphology. The majority are cubic CsPbBr_3_ perovskite crystals, between 3 nm and 10 nm wide. The rectangular shaped crystals are ascribed to CsPbBr_3_ nanoplatelets and might also be ascribed to cubic CsPbBr_3_ perovskites, which, due to local variations in composition, temperature and UV light exposure, self-assemble into nanocube superlattices [25,26,27]. Larger lateral size anisotropic 2D orthorhombic *Pnma* crystals are also seen, arising from superstructures of the CsPbBr_3_ building blocks. These blocks form linear chains of preferential crystallographic orientation and long-range order, that assemble into stacked columnar phases, creating larger rectangular structures [28]. The building blocks can also assemble creating non uniform structures. This is in accord with the medium and large size rectangular shapes and the irregular shapes seen in the STEM. The smaller rectangular structures have an average length of 12.5 nm, the larger structures measure between 25 nm and 35 nm. 

Large hexagonal structures also appear in the samples with the highest concentration of NCs and correspond to a phase transition from a 3D CsPbBr_3_ perovskite to a 0D Cs_4_PbBr_6_ perovskite. Besides being related to the local composition of the solution, these structures can be due to the presence of CsPbBr_3_ nanoplatelets, which agglomerate and originate a thermodynamically favored rhombohedral polymorph [28]. They display sides from 15 nm to 20 nm and heights up to 35 nm. All these different forms are seen in Figure 1d. 

### 3.2. The Impact of the Polymeric Encapsulation

Posteriorly to the synthesis, the obtained colloidal solution of NCs is deposited on top of float glass substrates of 2.5 × 3 cm, via spin coating. The deposited colloidal thin films were analyzed prior and post-deposition of the polymeric encapsulation of parylene type C, via CVD. 

#### 3.2.1. Optical Characterization of CsPbBr_3_ Nanocrystals and Thin Films with Parylene 

The visible light transmittance (VLT) decreases, from 85.9% to 77.7%, upon NCs deposition on the glass substrate and coating with the parylene thin film (Figure 2a). This is expected as the NCs and parylene layer absorb part of the incident radiation, decreasing light transmittance through the glass. However, 77.7% is still an acceptable transmittance.

The photoluminescence emission contour plot of the glass sample containing the luminescent NCs layer, and the polymeric encapsulation, does not show the presence of any contaminant species. A sharp emission profile is observed, peaking at 480 nm (2.58 eV) for an excitation wavelength of 290 nm (Figure 2b). No other emissive species are present. 

In comparison to the photoluminescence emission maximum of colloidal NCs (Figure 2c), there is a red shift in the emission maximum of the CsPbBr_3_ NCs thin film, from 500 nm (2.48 eV) to 509 nm (2.44 eV). However, the spectral line widths stay constant (Table 1), suggesting that the red shift may be related to the optical properties of the NC’s surrounding medium. For the CsPbBr_3_ NCs and parylene type C thin film, the emission maximum blue-shifts from 509 nm (2.44 eV) to 485 nm (2.56 eV). One possibility is that there is a change of the ligand passivating the surface of the NCs, thus blocking trap states and leading to a blue-shift in the emission wavelength [29]. 

The measured 2.46 eV band gap value of CsPbBr_3_ NCs dispersed in n-hexane (Table 1) is in line with the 2.24–2.38 eV values reported, and the slight increase is attributed to the presence of 2D nanoplatelets and rhombohedral Cs_4_PbBr_6_ NCs [30,31,32]. The measured 2.38 eV band gap of CsPbBr_3_ NCs thin film on a glass substrate matches literature values [33]. Upon deposition of the CsPbBr_3_ NCs thin film, the 1 μm polymer layer slightly increases to 2.54 eV the band gap of the final composite thin film, possibly due to alterations in the size of the CsPbBr_3_ NCs and the change of ligands at the surface of the nanocrystals [34]. This is in agreement with both the observed red-shift from solution to the thin film on glass substrate, and the observed blue-shift, from the solution to the CsPbBr_3_ NCs-parylene type C thin film (Figure 2c).

#### 3.2.2. XPS 

Appendix A depicts the XPS spectra of CsPbBr_3_ NCs thin film, CsPbBr_3_ NCs thin film with adhesion promotor and CsPbBr_3_ NCs-parylene type C thin film, all on glass substrates.

Figure 3a shows the Cs 3d_3/2_ (738.04 eV) and Cs 3d_5/2_ (724.01 eV) peaks witnessing the presence of Cs^+^ [35]. There is no shift upon the incorporation of the adhesion promoter and parylene type C coating.

Concerning the NCs thin film, the peaks fitted to the Pb 4f band (142.79 and 137.94 eV) are ascribed to Pb-oleate species (Figure 3b). There is a very slight peak shift with the incorporation of the adhesion promotor to CsPbBr_3_ NCs and two new peaks appear indicating the presence of Pb^2+^ [36] or PbCO_3_ [37]. As no peak at 138.8 eV is seen, the CsPbBr_3_ NCs thin film has no remains of the PbBr_2_ reactant [38]. After the encapsulation of the thin film with parylene type C the signal shifts to 143.07 eV (Pb 4f_5/2_) and 138.23 eV (Pb 4f_7/2_), meaning that there is a better Pb-Br binding. This suggests the thin film of CsPbBr_3_ NCs has V_Br_ vacancy defects that originate from intermediate energy levels in the band gap, with direct effect on the charge recombination dynamics, NCs stability and quantum yield [39]. The parylene type C deposition leads to the passivation of the oleylamine bromide terminations of the CsPbBr_3_ NCs, which bind to the Cl polymer terminations, reducing the number of V_Br_ defects, and, increasing the perovskite thin film quantum yield [40].

Considering Figure 3c, the peaks at 68.87 eV (Br 3d_3/2_) and 67.82 eV (Br 3d_5/2_) are attributed, respectively, to bromine atoms within the crystal and at the crystal’s surface [41].

Upon incorporation of the adhesion promotor on the CsPbBr_3_ NCs thin film, bands appear at 69.65 eV (Br 3d_3/2_) and 68.64 eV (Br 3d_5/2_) due to corner sharing of [PbBr_6_]^4−^ in CsPbBr_3_ [40]. As observed for the thin film of CsPbBr_3_ NCs encapsulated with parylene type C, the Br 3d_3/2_ and Br 3d_5/2_ peaks shift to higher binding energies, 69.12 eV and 68.08 eV, respectively. This, together with the shift towards higher binding energies of the peaks for the Pb 4f, indicates a change in the crystal structure, which may be related to the stabilization of the NCs in the form of Cs_4_PbBr_6_ [40]. 

The Cs/Pb/Br elemental ratios in Table 2 show that the CsPbBr_3_ NCs thin film is approximately stoichiometric. With the incorporation of the adhesion promotor, the Pb/Cs ratio of 1.81 and Br/Cs ratio of 1.64 indicate that the Br^-^ terminations partially bind to the silane. However, the parylene type C incorporation stabilizes the perovskite structure, with a partial binding of the Br^−^ terminations of CsPbBr_3_ NCs to the polymer Cl terminations, witnessed by a Br/Cs ratio of 2.15. 

The perovskite NCs crystal structure changes observed by XPS, upon the incorporation of the parylene encapsulation, parallel the photoluminescence emission blue-shift shift described in the optical characterization section. Hence, the observed optical shift is related with the tunability of the optical bandgap of the CsPbBr_3_ NCs, from the thin film to the final perovskite and parylene composite film.

### 3.3. Photovoltaic Cell Incorporation

To evaluate the ability of these luminescent thin films to work as Down-shifting Layers (DSL) and increase the performance of materials in energy saving applications, a system was developed with nc-Si:H PV cells (Figure 4a), in a transmission geometry. The characterization of the electrical properties was conducted both in the presence of NCs thin film, and after the deposition of the parylene encapsulation layer on the NCs containing samples.

#### Effect of the CsPbBr_3_ NCs-Parylene Type C Thin Film on PV Cells

The nc-Si:H PV cells were chosen since the emission maximum of the CsPbBr_3_ NCs is remarkably close to the maximum of the External Quantum Efficiency (EQE) of the cells. As seen in Figure 4b, and in Appendix A, the solar cell EQE increases in the absorption range of the NCs, that is, bellow 430 nm. The nc-Si:H PV cells show poor response in the UV due to front surface reflection and to parasitic absorption losses in the front contact. The enhancement factor for the EQE and absorption of the solar cells were calculated from the relative percentage differences between the uncoated PV cells and the coated PV cells with the CsPbBr_3_ NCs-parylene type C thin film. Appendix A show the plots of the enhancement factor for the EQE and absorption of the i, ii and, iii solar cell replicas, respectively. 

At 340 nm, an average enhancement factor of 58.36 ± 0.08% is obtained for the EQE of the PV cell with the incorporation of the CsPbBr_3_ NCs-parylene type C down-shifting layer. In the same wavelength, the absorption of the cell has an average enhancement factor of 5.76 ± 0.01%, in the presence of the thin film down-shifting layer. These results compare very well with those described in the work of Cao et al. [1], on a similar CsPbBr_3_ NCs thin film system for down-shifting in microcrystalline silicon solar cells. The paper demonstrated that, at 320 nm, there is an enhancement factor of 27% in the EQE and of 4% in the absorption of the solar cell. 

Our NCs absorb light in the UV region (up to 460 nm) and emit light at about 490 nm, which constitutes a down-shift of the UV light. Down-shifted light is emitted at all directions of space (much like light scattering) while part can in fact be re-absorbed by the NCs, since the Stokes shift is low. To deal with self-absorption effects, the NCs concentration was kept low to minimize this effect. In fact, light absorption by the NCs is estimated to be below 20% at the range where light is emitted (495 nm). Still, the NCs absorption at 460 nm may explain a slight decrease of the EQE observed in that spectral region.

For longer wavelengths, Figure 4b shows an EQE general enhancement above 460 nm and up to 610 nm. This effect was previously described in the work by Centeno et al. [12], on the application of parylene type C thin films in nanocrystalline silicon solar cells. These authors observed an increase in the external quantum efficiency of the PV cell in the 400 nm to 600 nm range, which is caused by the anti-reflective effect of the parylene coating. The latter results from the optical matching, consequence of the gradual refractive index increase: air (n = 1), parylene layer (n = 1.6), and the TCO layer of the solar cell (IZO, n = 1.9). This enhancement above 460 nm and up to 610 nm is confirmed by our results when NCs-parylene type C down-shifting layer is present.

The I–V curves of the PV cell replicas i, ii, and iii (Figure 5 and in Appendix A, respectively) show a very slight increase in both current and the voltage of the CsPbBr_3_ NCs-parylene type C coated cells, which translates in a slight increase on the value of the fill factor, and, consequently, on the overall power conversion efficiency of the solar cells (see Appendix A). The incorporation of the CsPbBr_3_ NCs-parylene type C down-shifting layer in the nc-Si:H PV cells increases, on average, 3.1% the short circuit current density and 5.6% the power conversion efficiency of the PV cells (the average values are shown in Table 3). The values for each replica can be consulted in Appendix A. Our results compare well with those described in literature by Meng et al. [4], on the application of CH_3_NH_3_PbBr_3_ quantum dots encapsulated in polyacrylonitrile as down-shifting layers in silicon solar cells. Meng et al. showed an increase of 2.07% and of 4.03% in the short circuit current density of microcrystalline and crystalline silicon solar cells, respectively. The power conversion efficiency was enhanced in 8.43% and 6.69% for the microcrystalline and crystalline silicon solar cells, respectively [42].

Table 4 shows that embedding CsPbBr_3_ NCs in the down-shifting layer increases, in average, by 6.02 ± 0.17% the absorption of light by the solar cells in the UV region, from 340 nm to 430 nm, which is the absorption range of the CsPbBr_3_ NCs. This translates into, in average, a 10.12 ± 0.75% increase of the external quantum efficiency. These enhancements were obtained from the ratios of the integrated curves, in percentage, from the spectral response data (external quantum efficiency and reflectance profiles). These measurements were performed, in dark conditions, without bias illumination, using a monochromator to gather the spectral response at each wavelength, in the 340 nm to 1100 nm spectral range.

Having the parylene type C coating as antireflective coating also increases the absorption of light and the external quantum efficiency of the solar cell in the 475–610 nm range, as previously explained in Centeno et al. [12] work. The developed down-shifting layer increases, in a reproducible way, by 8.13 ± 0.30% the absorption of light in the entire operation range of the silicon PV cells. 

## 4. Conclusions

A down-shifting layer consisting of CsPbBr_3_ nanocrystals encapsulated on a parylene type C matrix was developed and deposited on the surface of nc-Si:H photovoltaic cells to boost their low spectral response in the UV and visible range. The external quantum efficiency of the solar cells was improved by 10.12 ± 0.75%, in average, in the UV range, i.e., from 340 nm to 430 nm, caused by the down-shifting effect of the CsPbBr_3_ nanocrystals. Additionally, the anti-reflective effect of parylene type C also enhanced the solar cell’s performance, leading to an external quantum efficiency increase by 9.35 ± 1.13%, in average, in the visible range, from 460 nm to 610 nm. This is caused by the lower refractive index of parylene type C (n = 1.6) in comparison to the refractive index of the TCO layer beneath it (n = 1.9), which creates a gradual optical transition from the air to the active layer of the solar cell. The absorption of light increased by 8.13 ± 0.30%, in average, for the entire operation range of the solar cells, in the presence of the developed down-shifting layer. As a result, the short-circuit current density and the power conversion efficiency of the photovoltaic cells increased, respectively, 3.1% and 5.6% relative to the nc-Si:H PV cells without the CsPbBr_3_ NCs and parylene type C down-shifting layer. 

## Figures and Tables

**Figure 1 nanomaterials-13-00210-f001:**
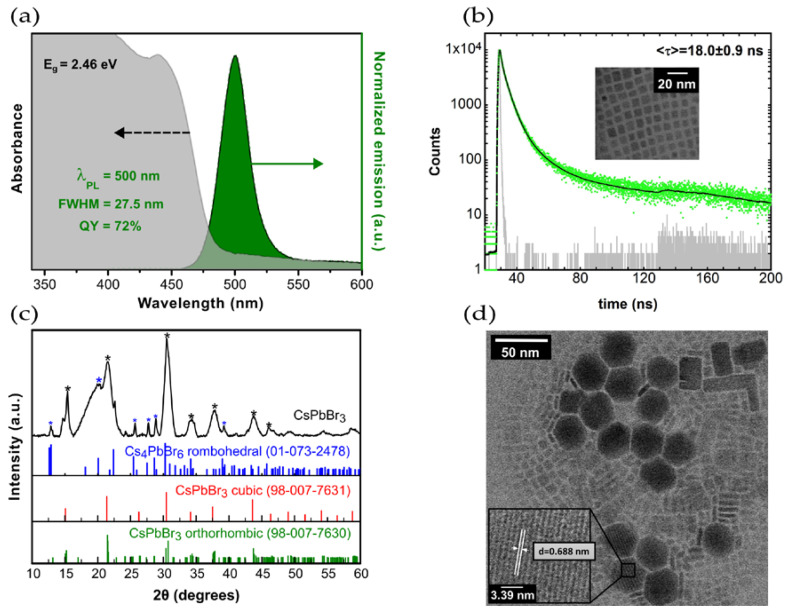
(**a**) Absorbance and photoluminescence emission spectra of CsPbBr_3_ nanocrystals in n-hexane; (**b**) photoluminescence decay of CsPbBr_3_ nanocrystals in n-hexane; (**c**) X-ray diffraction pattern of a drop cast from the higher concentration colloidal CsPbBr_3_ (pellet re-dispersed in 2.5 mL of hexane) on a quartz substrate, and the XRD reflections of cubic (ICSD ref. 98-007-7631) and orthorhombic (ICSD ref. 98-007-7630) CsPbBr_3_, and rhombohedral Cs_4_PbBr_6_ (ICSD ref. 01-073-2478); (**d**) STEM images of cesium lead perovskite nanocrystals obtained from the hot-injection synthesis showing the presence of three different phases; the inset depicts the rhombohedral 0D Cs_4_PbBr_6_ perovskite nanocrystals with clear lattice fringes and an estimated interplanar distance of 6.88 Å corresponding to the (110) plane.

**Figure 2 nanomaterials-13-00210-f002:**
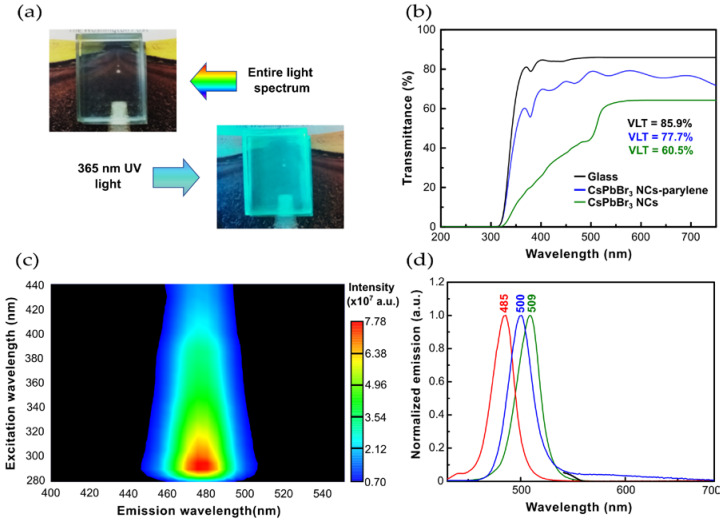
Encapsulation of CsPbBr_3_ NCs thin films with parylene type C: (**a**) coated glass sample under the entire light spectrum and 365 nm UV light; (**b**) impact on the light being transmitted through the glass (black line), when coated with CsPbBr_3_ NCs thin film (green line), and with the composite film of NCs-polymer (blue line); (**c**) contour plot of the photoluminescence emission intensity at different excitation wavelengths; (**d**) normalized emission spectra of CsPbBr_3_ NCs in solution (blue line), CsPbBr_3_ NCs thin film (green line), and CsPbBr_3_ NCs -parylene thin film (red line).

**Figure 3 nanomaterials-13-00210-f003:**
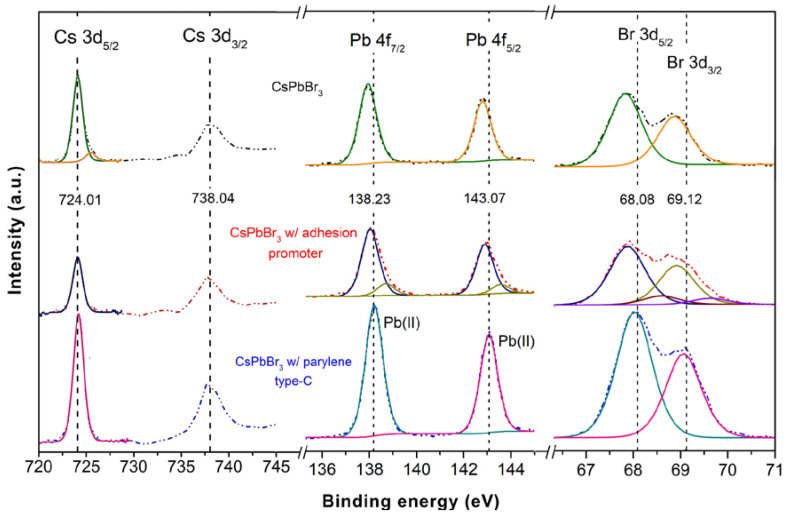
XPS Cs 3d, Pb 4f and Br 3d spectra of the CsPbBr_3_ NCs thin film (dashed black line), CsPbBr_3_ NCs thin film with the silane adhesion promotor (dashed red line), and CsPbBr_3_ NCs-parylene type C thin film (dashed blue line).

**Figure 4 nanomaterials-13-00210-f004:**
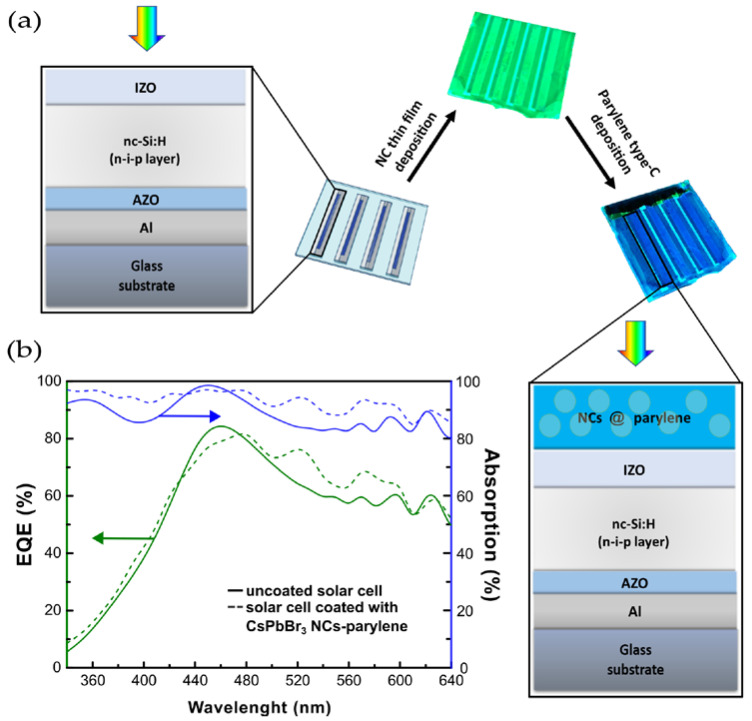
Photovoltaic cell integration: (**a**) layers that make up the uncoated nc-Si:H PV cells and the process of incorporation of the CsPbBr_3_ NCs-parylene type C thin film leading to the coated PV cell; (**b**) EQE (green lines) and absorption (blue lines) of the solar cell of replica ii before (solid line) and after (dashed line) the incorporation of the CsPbBr_3_ NCs-parylene type C thin film.

**Figure 5 nanomaterials-13-00210-f005:**
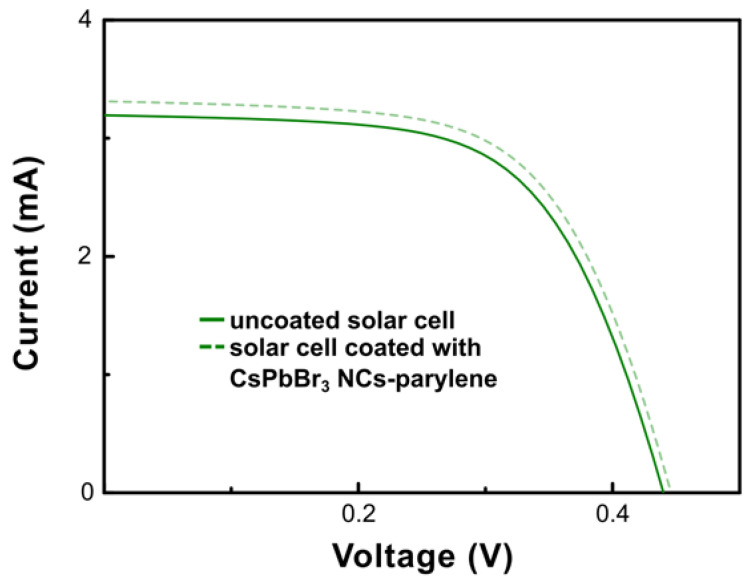
I–V curve of replica i before (solid green line) and after (dashed green line) the incorporation of the CsPbBr3 NCs-parylene type C thin film.

**Table 1 nanomaterials-13-00210-t001:** Optical bandgap, obtained from the Tauc plots, and the maximum wavelength and photoluminescence emission line width of the CsPbBr_3_ NCs in colloidal form and in thin film, with and without the parylene type C encapsulation.

Sample	λ_max_ (eV)	FWHM (eV)	E*_g_* (eV)
Colloidal CsPbBr_3_ NCs *	2.48	0.13	2.46
CsPbBr_3_ NCs thin film	2.44	0.12	2.38
CsPbBr_3_ NCs-parylene type C thin film	2.56	0.14	2.54

* synthesized pellet re-dispersed in 2.5 mL of n-hexane.

**Table 2 nanomaterials-13-00210-t002:** XPS elemental ratios.

Sample	Elemental Ratios of Cs/Pb/Br *
CsPbBr_3_ NCs thin film	1.00:1.10:2.80
CsPbBr_3_ NCs thin film with adhesion promotor	1.00:1.81:1.64
CsPbBr_3_ NCs-parylene type C thin film	1.00:1.25:2.15

* this quantification considers the equivalent homogeneous atomic fraction.

**Table 3 nanomaterials-13-00210-t003:** Average values of the measured photovoltaic cell parameters for replicas i, ii, and iii, when uncoated and coated with the CsPbBr3 NCs-parylene type C thin film: open-circuit voltage, short-circuit current, short-circuit current density, maximum current, maximum voltage, maximum power, fill factor and efficiency.

PV Cell	V_OC_ (V)	I_SC_ (mA)	J_SC_ (mA/cm^2^)	I_máx_ (mA)	V_máx_ (V)	P_máx_ (mW)	FF (%)	Efficiency (%)
uncoated	0.44 ± 0.01	3.20 ± 0.01	17.01 ± 0.04	2.72 ± 0.01	0.32 ± 0.01	0.87 ± 0.01	62.26 ± 0.45	4.67 ± 0.02
w/thin film	0.44 ± 0.01	3.28 ± 0.01	17.61 ± 0.13	2.82 ± 0.01	0.33 ± 0.01	0.92±0.01	63.53 ± 1.91	4.94 ± 0.06

**Table 4 nanomaterials-13-00210-t004:** Observed enhancement in the spectrally averaged absorption and external quantum efficiency of the PV cells, with CsPbBr_3_ NCs-parylene type C down-shifting layer relative to the uncoated references, along the UV range (from 340 nm to 430 nm) from the down-shifting effect of the NCs, and between 475–610 nm, from the anti-reflective properties of parylene type C.

	UV Range (340 nm to 430 nm)		Visible Range (475 nm to 610 nm)	
PV Cell Replica	Absorption Gain (%)	EQE Gain (%)	Absorption Gain (%)	EQE Gain (%)
i	6.09	10.07	8.22	7.77
ii	5.79	9.23	7.50	9.92
iii	6.18	11.06	8.94	10.36

## Data Availability

Not applicable.

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
