# Peer review of "Parylene-Sealed Perovskite Nanocrystals Down-Shifting Layer for Luminescent Spectral Matching in Thin Film Photovoltaics"

_nanomaterials, 2023, doi:10.3390/nano13010210_

Round 1

Reviewer 1 Report

In this paper, the authors applied the perovskite and parylene thin films as down-shifting layers to improve the performance of the nanocrystalline silicon photovoltaic solar cells. The perovskite nanocrystals thin films were spin coating and encapsulated with parylene type-C. The short circuit current density and power conversion efficiency were both observed to be slightly increased. There are some questions and concerns:

1.I-V curves of each cell are not presented in the paper. The authors need to provide the data.

2. The authors need to compare their data with other materials used for down-shifting layers

2. There are some typos, and grammar issues.

Overall, the authors need to revise the paper accordingly before it can be accepted.

Author Response

Reviewer 1

In this paper, the authors applied the perovskite and parylene thin films as down-shifting layers to improve the performance of the nanocrystalline silicon photovoltaic solar cells. The perovskite nanocrystals thin films were spin coating and encapsulated with parylene type-C. The short circuit current density and power conversion efficiency were both observed to be slightly increased. There are some questions and concerns:

1. I-V curves of each cell are not presented in the paper. The authors need to provide the data.

The authors acknowledge the reviewer for this important point. The I-V curves of solar cell replica i and it’s description, which was previously described in the supplementary materials, is now part of the main body of the paper (page 12). The I-V curves of solar cell replica ii and iii are present in Figures S7 and S8 of the supplementary materials.

2. The authors need to compare their data with other materials used for down-shifting layers

The authors thank the reviewer for this important guidance. There was a lack of data comparison with the results obtained in works from literature on down-shifting layers for silicon photovoltaic cells. This point was addressed and the obtained results were compared with ones from literature (in the new version those references are made in pages 10, 11 and 12 corresponding to references 1, 4, 12 and 42).

“At 340 nm, an average enhancement factor of 58.36±0.08% is obtained for the EQE of the PV cell with the incorporation of the CsPbBr3 NCs-parylene type C down-shifting layer. In the same wavelength, the absorption of the cell has an average enhancement factor of 5.76±0.01%, in the presence of the thin film down-shifting layer. These results compare very well with those described in the work of Cao et al.1, on a similar CsPbBr3 NCs thin film system for down-shifting in microcrystalline silicon solar cells. The paper demonstrated that, at 320 nm, there is an enhancement factor of 27% in the EQE and of 4% in the absorption of the solar cell.”

For longer wavelengths, Figure 4b shows an EQE general enhancement above 460 nm and up to 610 nm. This effect was previously described in the work by Centeno et al.12, on the application of parylene type C thin films in nanocrystalline silicon solar cells. These authors observed an increase in the external quantum efficiency of the PV cell in the 400 nm to 600 nm range, which is caused by the anti-reflective effect of the parylene coating.

The incorporation of the CsPbBr3 NCs-parylene type C down-shifting layer in the nc-Si:H PV cells increases, on average, 3.1% the short circuit current density and 5.6% the power conversion efficiency of the PV cells (the average values are shown in Table 3). The values for each replica can be consulted in Table S3 from the supplementary materials. Our results compare well with those described in literature by Meng et al.4, on the application of CH3NH3PbBr3 quantum dots encapsulated in polyacrylonitrile as down-shifting layers in silicon solar cells. Meng et al. showed an increase of 2.07% and of 4.03% in the short circuit current density of microcrystalline and crystalline silicon solar cells, respectively. The power conversion efficiency was enhanced in 8.43% and 6.69% for the microcrystalline and crystalline silicon solar cells, respectively.42"

3. There are some typos, and grammar issues.

The authors acknowledge the reviewer for this note. The paper was reviewed in order to eliminate these typos and grammar issues.

Overall, the authors need to revise the paper accordingly before it can be accepted.

The authors are grateful for the reviewer’s insights and have revised the paper accordingly to the comments made above. An effort was made in order to present the results more clearly and to expose the conclusions in a way that was supported by the obtained results.

Reviewer 2 Report

The authors have presented a significant amount of work investigating the crystal structure and emission spectra of UV illuminated nano-particles to be used as a down-conversion layer whereby UV light incident upon a microcrystalline solar is down-converted to a more efficiently converted to electrical power longer wavelength. The choice of nanoparticles investigated is novel. However the presentation and value of the approach is obscured by failure to adequately describe measurement conditions used to obtain stated efficiency gains. For example, fig. 4 is described as various cells (a, c, d) under various illuminations:

Table 4. Observed enhancement in the absorption and external quantum efficiency of a, c and d 499 cells, under 420 nm and between 475 nm-610 nm, from 

The problem is that the table only show one result and the illumination condition 420 nm or the wavelength range 475 to 610 nm could mean either two different measurements or one measurement under a monochromatic source and a broader range source - which is it? The nature of the 475 to 610 nm irradiance (photon flux as a function of wavelength is highly relevant)

Down conversion layers applied to solar cells is not a new concept. Reference to earlier work is needed here. For example,Solar Energy Materials and Solar Cells 108:241–245 DOI:10.1016/j.solmat.2012.08.011  and I am sure earlier work can be found with a little effort.

Every commercial solar cell employs an anti-reflection coating and here all of the comparisons meant to show efficiency gain used uncoated solar cells. This is not realistic and it make it difficult to assign gains to down-conversion as opposed to improvement in reflection reduction.

Finally the flaws in experimental verification through comparison stated above might be obscuring a possible theoretical flaw. Fluorescence does not preserve the incident beams direction. Given the relatively small (compared to silicon) refractive index of parylene (~ 1.66) a rather large fraction of the down converted light would be emitted in a direction away from the solar cell active layers. Furthermore, the states responsible for emitted the down converted light have the capacity to absorb the same wavelength bands from the incident solar spectrum, thereby, upon re-emission part of this irradiance would also be lost to the solar cell through emission into the loss cone angles towards the front of the cell. Whether or not actual gain results from down conversion depends critically on the amount of captured down-converted light from shorter wavelengths (by entering the active layers) versus the amount of light lost from the longer wavelengths bands due to large angle collateral scattering.

I think these various considerations can be addressed by describing the spectrums used for comparison measurements in greater detail. There is a reason these concepts have not been adopted by industry.

In summary please reference prior work on down conversion. Please provide more prior art context. Please provide meaningful comparisons to solar cells those with anti-reflection coatings, and detail the spectrums (photon fluxes) used to make comparison measurements on solar cells. There are a lot of interesting results.

Author Response

Reviewer 2

The authors have presented a significant amount of work investigating the crystal structure and emission spectra of UV illuminated nano-particles to be used as a down-conversion layer whereby UV light incident upon a microcrystalline solar is down-converted to a more efficiently converted to electrical power longer wavelength. The choice of nanoparticles investigated is novel. However the presentation and value of the approach is obscured by failure to adequately describe measurement conditions used to obtain stated efficiency gains. For example, fig. 4 is described as various cells (a, c, d) under various illuminations:

Table 4. Observed enhancement in the absorption and external quantum efficiency of a, c and d 499 cells, under 420 nm and between 475 nm-610 nm, from

The problem is that the table only show one result and the illumination condition 420 nm or the wavelength range 475 to 610 nm could mean either two different measurements or one measurement under a monochromatic source and a broader range source - which is it? The nature of the 475 to 610 nm irradiance (photon flux as a function of wavelength is highly relevant)

The authors thank the reviewer for this important point. The authors have made an effort to reformulate and to clarify the information described in Table 4 of page 13. The measurement conditions were better described in the last paragraph of page 4 The equipment uses a Newport-Oriel ® 6255, 150W, Ozone Free Xe ARC Lamp, which composes the illumination source of the equipment used. The spectral irradiance profile can be consulted in the method details section of the supplementary materials.”  and in the first paragraph of page 12. The data from this table was acquired from the external quantum efficiency and reflectance profiles from the spectral response measurements. The absorption profiles for the PV cell replicas i, ii, and iii were gathered from the reflectance data. The enhancement values presented arose from the integration of both the external quantum efficiency and absorption profiles of the solar cell replicas, from the 340 nm to 430 nm range, which is the absorption range of the CsPbBr3 NCs. The percentual ratios were acquired from these results, between the integrated value for the coated solar cell replica with the CsPbBr3 NCs-parylene type C thin film and the integrated value for the uncoated solar cell replica. The alterations mentioned above can be consulted in the new version of the paper in the first paragraph of page 10 and in the first paragraph of page 12.

“The enhancement factor for the EQE and absorption of the solar cells were calculated from the relative percentage differences between the uncoated PV cells and the coated PV cells with the CsPbBr3 NCs-parylene type C thin film. Figures S4, S5 and, S6 from the supplementary materials show the plots of the enhancement factor for the EQE and absorption of the i, ii and, iii solar cell replicas, respectively.

Table 4 shows that embedding CsPbBr3 NCs in the down-shifting layer increases, in average, by 6.02±0.17% the absorption of light by the solar cells at wavelengths in the UV region, from 340 nm to 430 nm, which is the absorption range of the CsPbBr3 NCs. This, translates into, in average, a 10.12±0.75% increase of the external quantum efficiency. These enhancements were obtained from the ratios of the integrated curves, in percentage, from the spectral response data (external quantum efficiency and reflectance profiles). These measurements were performed, in dark conditions, without bias illumination, using a monochromator to gather the spectral response at each wavelength, in the 340 nm to 1100 nm spectral range.

Down conversion layers applied to solar cells is not a new concept. Reference to earlier work is needed here. For example,Solar Energy Materials and Solar Cells 108:241–245 DOI:10.1016/j.solmat.2012.08.011  and I am sure earlier work can be found with a little effort.

The authors thank the reviewer for this important point. There was a lack of data comparison with the results obtained in works from literature on down-shifting layers for silicon photovoltaic cells. This point was addressed and the obtained results were compared with ones from literature (in the new version those references are made in pages 10, 11 and 12 corresponding to references 1, 4, 12 and 42).

“At 340 nm, an average enhancement factor of 58.36±0.08% is obtained for the EQE of the PV cell with the incorporation of the CsPbBr3 NCs-parylene type C down-shifting layer. In the same wavelength, the absorption of the cell has an average enhancement factor of 5.76±0.01%, in the presence of the thin film down-shifting layer. These results compare very well with those described in the work of Cao et al.1, on a similar CsPbBr3 NCs thin film system for down-shifting in microcrystalline silicon solar cells. The paper demonstrated that, at 320 nm, there is an enhancement factor of 27% in the EQE and of 4% in the absorption of the solar cell.”

For longer wavelengths, Figure 4b shows an EQE general enhancement above 460 nm and up to 610 nm. This effect was previously described in the work by Centeno et al.12, on the application of parylene type C thin films in nanocrystalline silicon solar cells. These authors observed an increase in the external quantum efficiency of the PV cell in the 400 nm to 600 nm range, which is caused by the anti-reflective effect of the parylene coating.

The incorporation of the CsPbBr3 NCs-parylene type C down-shifting layer in the nc-Si:H PV cells increases, on average, 3.1% the short circuit current density and 5.6% the power conversion efficiency of the PV cells (the average values are shown in Table 3). The values for each replica can be consulted in Table S3 from the supplementary materials. Our results compare well with those described in literature by Meng et al.4, on the application of CH3NH3PbBr3 quantum dots encapsulated in polyacrylonitrile as down-shifting layers in silicon solar cells. Meng et al. showed an increase of 2.07% and of 4.03% in the short circuit current density of microcrystalline and crystalline silicon solar cells, respectively. The power conversion efficiency was enhanced in 8.43% and 6.69% for the microcrystalline and crystalline silicon solar cells, respectively.42

Every commercial solar cell employs an anti-reflection coating and here all of the comparisons meant to show efficiency gain used uncoated solar cells. This is not realistic and it make it difficult to assign gains to down-conversion as opposed to improvement in reflection reduction.

The authors acknowledge the reviewer for this comment and included a reference (reference 12, doi:10.1002/admi.202000264) to a previously developed work, which employs parylene type C thin film coatings on top of nanocrystalline solar cells (which are deposited in a similar manner to those used in our work- the same equipment, materials and methods, as a standard procedure of the laboratory). This referenced work demonstrates that the gains from the improvement in reflection reduction occur only in the 400 nm to 600 nm range. There is no evidence, in the referenced work, of the improvement in efficiency of the solar cells in the UV region, as seen in Figure 5 from Centeno et. al. paper, which is where the CsPbBr3 NCs, present in the down-shifting layer, will have an important role in reducing the light being reflected from the front contact of the photovoltaic cells and boosting its performance, in the region were these nanocrystalline silicon solar cells show the poorest response. These alterations were introduced in the last paragraph of page 10.

“For longer wavelengths, Figure 4b shows an EQE general enhancement above 460 nm and up to 610 nm. This effect was previously described in the work by Centeno et al.12, on the application of parylene type C thin films in nanocrystalline silicon solar cells. These authors observed an increase in the external quantum efficiency of the PV cell in the 400 nm to 600 nm range, which is caused by the anti-reflective effect of the parylene coating. The latter results from the optical matching, consequence of the gradual refractive index increase: air (n=1), parylene layer (n=1.6), and the TCO layer of the solar cell (IZO, n=1.9). This enhancement above 460 nm and up to 610 nm is confirmed by our results when NCs-parylene type C down-shifting layer is present.

Finally the flaws in experimental verification through comparison stated above might be obscuring a possible theoretical flaw. Fluorescence does not preserve the incident beams direction. Given the relatively small (compared to silicon) refractive index of parylene (~ 1.66) a rather large fraction of the down converted light would be emitted in a direction away from the solar cell active layers. Furthermore, the states responsible for emitted the down converted light have the capacity to absorb the same wavelength bands from the incident solar spectrum, thereby, upon re-emission part of this irradiance would also be lost to the solar cell through emission into the loss cone angles towards the front of the cell. Whether or not actual gain results from down conversion depends critically on the amount of captured down-converted light from shorter wavelengths (by entering the active layers) versus the amount of light lost from the longer wavelengths bands due to large angle collateral scattering.

The authors agree with the points raised by the reviewer, however, conscious efforts were made to try to avoid the issue of re-absorption of light, by employing low enough concentrations of colloidal CsPbBr3 NCs in the developed thin films. Besides the Parylene type C layer is not in direct contact with the silicon, but rather in contact with the TCO. These point are addressed  in the last section of the first paragraph from page 10 and from page 13. Additionally, despite all the theoretical losses, we were still able to observe gains in the UV region of operation of the solar cells, meaning that the overall balance still favours an enhancement in the performance of the PV component described in this work. A complete theoretical description of this technology was not within the framework presented here. Still, these are laboratory results, and any scale-up would require a lot of additional work. How to avoid light leakage from the films must be addressed in the future.

Our NCs absorb light in the UV region (up to 460 nm) and emit light at about 490 nm, which constitutes a down-shift of the UV light. Down-shifted light is emitted at all directions of space (much like light scattering) while part can in fact be re-absorbed by the NCs, since the Stokes shift is low. To deal with self-absorption effects, the NCs concentration was kept low to minimize this effect. In fact, light absorption by the NCs is estimated to be below 20% at the range where light is emitted (495 nm). Still, the NCs absorption at 460 nm may explain a slight decrease of the EQE observed in that spectral region.

“Additionally, the anti-reflective effect of parylene type C also enhanced the solar cell’s performance, leading to an external quantum efficiency increase by 9.35±1.13%, in average, in the visible range, from 460 nm to 610 nm. This is caused by the lower refractive index of parylene type C (n=1.6) in comparison to the refractive index of the TCO layer beneath it (n=1.9), which creates a gradual optical transition from the air to the active layer of the solar cell.”

I think these various considerations can be addressed by describing the spectrums used for comparison measurements in greater detail. There is a reason these concepts have not been adopted by industry.

The authors are grateful for the reviewer’s insights. To clarify the obtained results and conclusions, the authors have described and added the enhancement factor data to the paper (in the first paragraph of page 10 and in the first paragraph of page 12, which refers to the information presented in Table 4), along with the correspondent profiles for the enhancement factor calculated for the external quantum efficiency and absorption of the solar cell replicas i, ii, and iii, which can be consulted in Figures S4, S5 and S6 from the supplementary materials. The irradiance spectra of the illumination source can also be consulted in the method details section of the supplementary materials.

“The enhancement factor for the EQE and absorption of the solar cells were calculated from the relative percentage differences between the uncoated PV cells and the coated PV cells with the CsPbBr3 NCs-parylene type C thin film. Figures S4, S5 and, S6 from the supplementary materials show the plots of the enhancement factor for the EQE and absorption of the i, ii and, iii solar cell replicas, respectively.

Table 4 shows that embedding CsPbBr3 NCs in the down-shifting layer increases, in average, by 6.02±0.17% the absorption of light by the solar cells at wavelengths in the UV region, from 340 nm to 430 nm, which is the absorption range of the CsPbBr3 NCs. This, translates into, in average, a 10.12±0.75% increase of the external quantum efficiency. These enhancements were obtained from the ratios of the integrated curves, in percentage, from the spectral response data (external quantum efficiency and reflectance profiles). These measurements were performed, in dark conditions, without bias illumination, using a monochromator to gather the spectral response at each wavelength, in the 340 nm to 1100 nm spectral range.

In summary please reference prior work on down conversion. Please provide more prior art context. Please provide meaningful comparisons to solar cells those with anti-reflection coatings, and detail the spectrums (photon fluxes) used to make comparison measurements on solar cells. There are a lot of interesting results.

New references and results comparisons to works present in literature were integrated in the paper. The paper published on previous work employing the parylene type C as anti-reflection coating on nanocrystalline solar cells was referenced and used to compare, clarify and distinguish the region of operation of the anti-reflective coating from the region of operation of the down-shifting layer. Additionally, further data and profiles that back up the results and conclusions from this work were added to the paper and to the supplementary materials section.

Round 2

Reviewer 2 Report

The revised manuscript is improved. The might be better placed within the context of what gain can be expected in a terrestrial solar cell where quantum efficiency is considered against established solar spectrum. Nonetheless, I think the manuscript should proceed to publication. I have outlined my calculation of expected gain under AM1.5 illumination below for the authors consideration (revision is not required).

Does the gain in UV (425nm) response outweigh the loss in response at the emission wavelength (495 nm) due to large angle scattering (absorption and re-emission) when the solar irradiance is taking into consideration? I do not think this has been demonstrated. A reasonable solar cell (silicon or micro crystalline) should have a peak internal quantum efficiency of ~ 100% at 495 nm. At 425 nm the internal quantum efficiency should easily be 65%. Therefore, the combined currents from 425 nm and 495 nm currents (where the 495 nm current is taken as unity for exercise) and adjusting for the difference in photon energy (it takes fewer 425 nm photons to make the irradiant energy found at 425 nm) yields a combined current (425 and 495 nm) that is 37%greater than the 495 nm current alone:

 [

Since the global AM 1.5 irradiance standard adopted by NREL indicates that the irradiant power at 495 nm is 1.5 times greater than that at 425 nm. Therefore, the photon flux at 425 nm is 67% that at 495nm. Now adjusting for the higher energy of 425 nmphotons carrying greater energy per photon relative to 495 nm photons further reduces the photon flux (and therefore the expected current gain due 425 nm photons) by the ratio 425/495. What can be gained by down conversion?

Assuming both photons at 425 and 495 nm are both subject to back scattering loss through a lose-cone corresponding to a a refractive index of 1.66 about 10% of the incident irradiation at both wavelengths is lost leaving 90% available for current generation. 

So, is there really a 3.6% gain on the band of light down-converted? It would seem unlikely. While the down conversion layer may provide some index matching it would still need one or more quarter-wavelength anti-reflection coating(s) and the energy loss cone calculation (10% loss) used above only applies to planar down-conversion layers, losses increase significantly when the encapsulant layer flows conformally over the down-conversion particles. Also, when the entire solar spectrum is considered the 400 to 450 nm band contributes less than ~5% of the total current of a silicon-based solar cell under AM1.5 irradiance indicating the net efficiency gain under the most favorable scattering loss assumption (planar air/encapsulant layer interface) would be ~0.18%.

Author Response

Reviewer 2

The revised manuscript is improved. The might be better placed within the context of what gain can be expected in a terrestrial solar cell where quantum efficiency is considered against established solar spectrum. Nonetheless, I think the manuscript should proceed to publication. I have outlined my calculation of expected gain under AM1.5 illumination below for the authors consideration (revision is not required).

Does the gain in UV (425nm) response outweigh the loss in response at the emission wavelength (495 nm) due to large angle scattering (absorption and re-emission) when the solar irradiance is taking into consideration? I do not think this has been demonstrated. A reasonable solar cell (silicon or micro crystalline) should have a peak internal quantum efficiency of ~ 100% at 495 nm. At 425 nm the internal quantum efficiency should easily be 65%. Therefore, the combined currents from 425 nm and 495 nm currents (where the 495 nm current is taken as unity for exercise) and adjusting for the difference in photon energy (it takes fewer 425 nm photons to make the irradiant energy found at 425 nm) yields a combined current (425 and 495 nm) that is 37%greater than the 495 nm current alone:

The authors acknowledge the points made by the reviewer. The total EQE gain, for the 340 nm to 1100 nm spectral range, has a 1.07±1.85% measured increase, in average. In these three replicas there is an indication from EQE results of a slight gain as a result of QDs. However, under the AM1.5 irradiation, the I-V curves show an average relative increase of 5.6±1.1% in efficiency. This includes the contributions of the QDs an the antireflection of parylene type C, which we cannot differentiate in the I-V curve measurement.

Since the global AM 1.5 irradiance standard adopted by NREL indicates that the irradiant power at 495 nm is 1.5 times greater than that at 425 nm. Therefore, the photon flux at 425 nm is 67% that at 495nm. Now adjusting for the higher energy of 425 nmphotons carrying greater energy per photon relative to 495 nm photons further reduces the photon flux (and therefore the expected current gain due 425 nm photons) by the ratio 425/495. What can be gained by down conversion?

The down conversion translates a gain in the UV region, were the light being absorbed by the CsPbBr3 is simultaneously being emitted as visible light, which, in turn, is absorbed by the active layers of the PV cell causing an increase in the current being generated and, consequently, an increase in the solar cell’s EQE in the UV region, where the effect occurs.

Assuming both photons at 425 and 495 nm are both subject to back scattering loss through a lose-cone corresponding to a a refractive index of 1.66 about 10% of the incident irradiation at both wavelengths is lost leaving 90% available for current generation.  

So, is there really a 3.6% gain on the band of light down-converted? It would seem unlikely. While the down conversion layer may provide some index matching it would still need one or more quarter-wavelength anti-reflection coating(s) and the energy loss cone calculation (10% loss) used above only applies to planar down-conversion layers, losses increase significantly when the encapsulant layer flows conformally over the down-conversion particles. Also, when the entire solar spectrum is considered the 400 to 450 nm band contributes less than ~5% of the total current of a silicon-based solar cell under AM1.5 irradiance indicating the net efficiency gain under the most favorable scattering loss assumption (planar air/encapsulant layer interface) would be ~0.18%.

These are experimental results. Further model developing of these types of devices, useful for the understanding of the mechanisms pointed up by the reviewer, should be addressed in future studies.